# The Moderating Effects of Self-Care on the Relationships between Perceived Stress, Job Burnout and Retention Intention in Clinical Nurses

**DOI:** 10.3390/healthcare11131870

**Published:** 2023-06-27

**Authors:** Seung-Hee Lee, Min-Ho Joo

**Affiliations:** 1Post Anesthesia Care Unit 2 (PACU 2), Seoul National University Hospital, 101 Daehak-ro, Jongno-gu, Seoul 03080, Republic of Korea; 17635@snuh.org; 2Department of Educational Technology, Konkuk University, Seoul 05029, Republic of Korea

**Keywords:** burnout, nurses, retention, self-care, stress

## Abstract

This study determines the importance of nurses’ self-care by identifying its effects as a moderating variable on the relationships between perceived stress, job burnout, and retention intention in clinical nurses. Hence, 174 clinical nurses who worked at two university hospitals and one general hospital located in Seoul, South Korea, participated in this study. As the hospitals required the use of recruitment notices, convenience sampling methods were adopted to recruit volunteers. The data were collected using the perceived stress scale, the burnout assessment tool, the nurse retention index, and the self-care assessment worksheet. Descriptive statistics were used to analyze the general characteristics of participants, and *t*-test and analysis of variance were conducted for comparisons. Moderated multiple regression was conducted to verify the moderating effects of self-care on the relationships between perceived stress and retention intention and between job burnout and retention intention. The results revealed that the effect of perceived stress on retention intention and the moderating effect of self-care on the relationship between perceived stress and retention intention were not significant, whereas job burnout had a direct effect on retention intention, and self-care had a positive moderating effect on job burnout. Therefore, it is necessary to consider an instructional program on the importance of self-care and conduct campaign activities at the organizational level. Moreover, various support structures should be provided at the organizational level such that nurses can reduce their levels of job burnout.

## 1. Introduction

Due to the recent prolonged coronavirus disease 2019 (COVID-19) pandemic, nurses, who were at the forefront during the quarantine, have been continuously exposed to infection and safety threats and have experienced extreme stress, which has been identified as a cause of burnout [1]. In addition, a study conducted on healthcare workers in 33 countries demonstrated that 51% of the participants experienced burnout related to COVID-19; the factors that greatly affected exhaustion included high work stress, heavy workload, time pressures, and lack of hospital support [2]. The current ratio of nurses in South Korea is 8.4 per 1000 of the population, which remains lower than the Organisation for Economic Co-operation and Development (OECD) average of 10.0 [3], and the issue of burnout among nurses is becoming more serious.

People with long-term jobs commonly experience job burnout, a negative behavioral phenomenon caused by a combination of emotional exhaustion, depersonalization, and decreased individual achievement [4]. In particular, nurses are prone to burnout because they experience high job stress during the process of providing nursing services related to the patient’s health life and death. In turn, resilience is reduced, and, if they continue to experience job stress, burnout occurs [5,6]. As such, burnout negatively affects the physical and mental health of nurses while causing negative occupational attitudes, which impairs the quality of nursing services, increases nursing errors, and reduces job satisfaction, resulting in high turnover intention [7].

A high turnover rate may increase the workload of incumbent nurses, increase stress, and reduce morale, causing serious problems in nursing performance [8]. As a result, effective nursing activities become difficult; job satisfaction decreases, and some incumbent nurses consider changing jobs, which can lead to a vicious cycle [9].

Perceived stress refers to the degree of stress at which the external environment exceeds the resources possessed by an individual and negatively affects physical and psychological wellbeing [10]. This degree can vary depending on how individuals interpret and accept external stimuli [11]. For example, even in the same stressful situation, some nurses respond appropriately and experience less burnout, whereas others experience exhaustion [12]. This stress is correlated with retention intention [13], and when similar problems are repeated and unresolved, or when serious work conflicts emerge, nurses’ retention intention can be reduced [14].

Self-care is another factor associated with burnout, stress, and retention intention. It refers to self-management activities and strategies in physical, psychological, emotional, spiritual, and professional or workplace categories and to work–life balance. No study was located on the effects of self-care upon the relationship between stress, burnout, and retention intention; however, the results of a study on the effects of nurses’ self-care on empathy fatigue demonstrated that the lesser the self-care, the higher the empathy fatigue [15]. Regarding this finding, previous studies have identified that nurses may neglect their self-care activities due to shift changes and busy work environments [15].

No theoretical framework or structural model includes the variables mentioned above. A synthesis of the research on the causal relationships related to these variables indicates the following: perceived stress and job burnout have negative effects on retention intention. The root causes of perceived stress and job burnout can be attributed to the workplace. However, different individuals, even when experiencing the same external stimuli, can manifest perceived stress and job burnout differently. Therefore, in addition to an organizational approach, individual self-care, which allows for individual control, is important. Hence, based on the causal relationships of these variables, this study explores the mediating role of self-care in the relationships between perceived stress, job burnout, and retention intention in clinical nurses.

Furthermore, several studies report that retention intention may vary depending on the degree of stress and burnout; however, previous studies providing adequate exploration of practical and applicable solutions are still lacking [13,16]. Some studies have been conducted on patients’ self-care [17,18]; however, studies on nurses’ self-care, who are key professional personnel in medical services, are insufficient. Therefore, this study aims to determine the importance of nurses’ self-care by identifying its effects as a moderating variable in the relationship between perceived stress, job burnout, and retention intention in clinical nurses. The detailed objectives of this study are as follows:Assessing the participants’ perceived stress, job burnout, retention intention, and self-management levelsIdentifying the differences in perceived stress, job burnout, retention intention, and degree of self-care, depending on the general characteristics of the participantsIdentifying the correlations among the participants’ perception of stress, job burnout, retention intention, and self-careInvestigating the moderating effects of self-care on the relationship between stress and retention intention and between job burnout and retention intention.

The hypotheses established in this study are as follows:Self-care has a moderating effect on the relationship between perceived stress and retention intention in clinical nurses.Self-care has a moderating effect on the relationship between job burnout and retention intention in clinical nurses.

## 2. Research Methods

### 2.1. Research Design

This study examines the effects of nurses’ perceived stress and job burnout on retention intention and identifies the moderating effects of self-care on the relationship between perceived stress, job burnout, and retention intention.

### 2.2. Participants

The study participants comprised nurses who worked at two major university hospitals and one general hospital located in Seoul, South Korea. The hospitals allowed us to collect data and required the use of recruitment notices; therefore, convenience sampling was used to recruit volunteers. To verify the moderating effects of self-care on the relationship between stress and retention intention and between job burnout and retention intention, G*Power 3.1, a sample size calculation program, was used to identify the minimum sample size. The effect size was 0.15; the significance level was 0.05; and the power was 0.95; as a result of applying three predictors, including a moderating variable, a minimum sample size of 119 people was calculated for multiple regression analysis. In this study, a total of 176 samples were collected, and 174 samples were used for the analysis, with 2 responses that were not applicable to clinical nurses excluded. Nurses above the manager level were excluded from the study because they were not categorized as clinical nurses in Korea. The overall response rate was 98.86%.

### 2.3. Survey Questionnaires

The questionnaire used in this study included 10 questions on perceived stress, 22 on job burnout, 6 on retention intention, and 65 on self-care.

#### 2.3.1. Perceived Stress

The perceived stress scale (PSS) used in this study was developed by Cohen et al. in 1983 and was modified by Cohen et al. in 1988 [19]. This questionnaire was translated and used by Lee et al. [20] with the written consent of the developer. The reliability of the questionnaire developed by Cohen et al. was Cronbach’s α = 0.90. The PSS consisted of ten questions, and scores were applied on a 5-point Likert scale (0, never; 1, rarely; 2, sometimes; 3, frequently; and 4, very often). Among the questions, questions 4, 5, 7, and 8 were reversely calculated during data analysis. The total score ranged from 0 to 40, with a high score indicating a high degree of perceived stress. The reliability of the Korean version of the stress scale was α = 0.82 and that of this study was α = 0.84.

#### 2.3.2. Job Burnout

The Korean version of the burnout assessment tool (K-BAT), adapted by Cho [21] from the burnout assessment tool (BAT) developed by Schaufeli et al. [22], was the job burnout questionnaire used in this study. K-BAT consisted of 22 questions and a 5-point Likert scale (1, never; 2, rarely; 3, sometimes; 4, often; and 5, always) for four subfactors. The subfactors were eight questions for exhaustion, four for mental distance, five for cognitive impairment, and five for emotional impairment. A high score indicated a high level of job burnout. The reliability of K-BAT was within acceptable limits with Cronbach’s alpha ranging from 0.90 to 0.92 for the subfactors. The reliability of the questionnaire in this study was exhaustion (α = 0.87), mental distance (α = 0.86), cognitive impairment (α = 0.89), and emotional impairment (α = 0.91). The overall reliability was α = 0.91.

#### 2.3.3. Retention Intention

Retention intention was measured using a questionnaire that translated the nurses’ retention index developed by Cowin [23] into Korean [24]. The questionnaire consisted of six questions, and answers were scored on an 8-point Likert scale, with a high score indicating high retention intention (1, definitely false; 2, false; 3, mostly false; 4, more false than true; 5, more true than false; 6, mostly true; 7, true; and 8, definitely true). Of the six questions, the third and sixth were reverse-ordered questions; thus, they were reversely calculated during data analysis. The reliability of the Korean version questionnaire was α = 0.89 [24] and that of this study was α = 0.91.

#### 2.3.4. Self-Care

Self-care was measured using a questionnaire that translated the self-care assessment worksheet (SCAW) developed by Saakvitne et al. [25] into Korean [16]. The SCAW consisted of 65 questions scored on a 5-point Likert scale (1, it never occurred to me; 2, never; 3, rarely; 4, occasionally; and 5, frequently) for six subfactors. The subfactors consisted of 14 questions for physical self-care, 12 for psychological self-care, 10 for emotional self-care, 16 for spiritual self-care, 11 for workplace or professional self-care, and 2 for balance, with a high score indicating better self-care. The reliability of the Korean version of SCAW was within acceptable limits with Cronbach’s alpha ranging from 0.84 to 0.93 for the subfactors. The reliability of the questionnaire in this study was α = 0.81, 0.87, 0.80, 0.91, 0.84, and 0.79 for physical, psychological, emotional, spiritual, workplace or professional self-care, and work–life balance, respectively. The overall reliability was α = 0.95.

### 2.4. Data Collection and Ethical Considerations

The data collection period for this study was from 27 January to 24 February 2023. Ethics approval was received from the Institutional Review Board (IRB) of the university hospital located in Seoul, South Korea to protect the rights and interests of the study participants (IRB Number H-2301-068-1394). Before data collection, the study purpose and method were explained to the department head, and a recruitment notice was posted on the ward bulletin board. Then, the survey contents were explained based on the research description, and the consent of those who wanted to participate in the study was obtained. Participants completed an online questionnaire using a URL or QR code in the notice of participation.

### 2.5. Data Analysis

The following analysis was conducted using the SPSS 25.0 program for the collected data:Descriptive statistics analyzed the degree of perceived stress, job burnout, self-care, and retention intention of clinical nurses.The *t*-test and analysis of variance were used to compare and analyze the differences in perceived stress, job burnout, self-care, and retention intention based on the general characteristics of clinical nurses.Moderated multiple regression analysis was performed to determine the moderating effects of self-care on the relationship between perceived stress, job burnout, and retention intention of clinical nurses.

## 3. Results

### 3.1. Descriptive Analysis

The participants’ general characteristics were gender, age, clinical work experience, marital status, religion, department, and hobbies (Table 1). Among these characteristics, age was classified into <29, 30–34, 35–39, and >40 years. Clinical work experience was divided into beginner (<1 year), advanced beginner (2–3 years), competent (4–6 years), and proficient (>7 years) stages based on their career [26]. The total number of participants in this study was 174.

The descriptive analysis of variables is reported in Table 2. The mean for perceived stress was 1.83, which was lower than the median value of 2. Meanwhile, the means for job burnout, self-care, and retention intention were 2.61, 2.73, and 5.05, respectively. These results indicate that the mean values for job burnout, self-care, and retention intention were higher than their respective median values. The skewness and kurtosis values representing the data for each variable were normally distributed.

### 3.2. Comparison of Perceived Stress, Job Burnout, Self-Care, and Retention Intention Based on General Characteristics

To determine whether nurses’ perceived stress, job burnout, self-care, and retention intention differed based on their general characteristics, their retention intention was compared and analyzed, focusing on gender, age, clinical work experience, marital status, religion, department, and hobbies (Table 3). Before the comparison, a Shapiro–Wilk test was conducted to identify the normality of data, producing a value of 0.986 (*p* = 0.069). Because the *p*-value was greater than 0.05, the data satisfied the assumption of normality. A *t*-test and analysis of variance were conducted for the comparisons. Regarding the difference in perceived stress by gender, the perceived stress of male nurses was found to be higher than that of female nurses, whereas the job burnout of female nurses was identified to be significantly higher than that of male nurses. No significant difference was observed in self-care and retention intention by gender. Moreover, no differences were observed in perceived stress, job burnout, self-care, and retention intention by age, clinical experience, marital status, religious status, and working department. Nurses without hobbies had significantly higher perceived stress and job burnout than those with hobbies. In addition, nurses with hobbies had significantly higher self-care rates than those without.

### 3.3. Correlations among Perceived Stress, Job Burnout, Self-Care, and Retention Intention

Examining the correlations among perceived stress, job burnout, self-care, and retention intention reveals that perceived stress was significantly positively correlated with job burnout (r = 0.587, *p* < 0.01) but negatively with self-care (r = −0.305, *p* < 0.01) and retention intention (r = −0.272, *p* < 0.01). Job burnout was also significantly negatively correlated with self-care (r = −0.287, *p* < 0.01) and retention intention (r = −0.424, *p* < 0.01). Self-care was significantly positively correlated with retention intention (r = 0.321, *p* < 0.01) (Table 4).

### 3.4. Moderating Effects of Self-Care on the Relationships between Perceived Stress, Job Burnout, and Retention Intention

Before performing moderated multiple regression analysis, this study identified whether the assumptions of multiple regression were satisfied. The Durbin–Watson value for independent verification was 1.75, which is close to 2, confirming the absence of autocorrelation. Multicollinearity was checked with tolerance (0.758–0.921) and VIF (1.086–1.319), and no multicollinearity problem was observed. After checking the assumptions of multiple regression, moderated multiple regression was performed to verify the moderating effects of self-care on the effects of perceived stress and job burnout on retention intention. Because of the hypothesis model verification, level 1, the direct effect of independent variables; level 2, the direct effect of moderating variables; and level 3, the interaction effects of independent and moderating variables, were all statistically significant (Table 5). The *R^2^* of levels 1, 2, and 3 was 0.106, 0.158, and 0.183, respectively. The extent of *R^2^* changes from level 1 to 2 was statistically significant (*R^2^* change = 0.052, *p* = 0.001), whereas that from level 2 to 3 was not (*R^2^* change = 0.024, *p* = 0.085). Based on the result of level 3 (F = 7.515, *p* < 0.001), perceived stress and the interaction between perceived stress and self-care were not significantly associated with retention intention; thus, the effects of perceived stress were not significant among the independent variables set in the hypothesis model.

Therefore, perceived stress was eliminated from the hypothesis model and set as a modified model. Then, the moderating effects of self-care on job burnout on retention intention were identified again. With the modified model verification, the *R^2^* of levels 1, 2, and 3 were 0.098, 0.157, and 0.177, respectively. The extent of *R^2^* changes from levels 1 to 2 was significant (*R^2^* change = 0.058, *p* = 0.001), as was that from levels 2 to 3 (*R^2^* change = 0.020, *p* = 0.043). Based on the result of level 3 (F = 12.191, *p* < 0.001), job burnout (*β* = −0.245, *p* = 0.001), self-care (*β* = 0.282, *p* < 0.001), and the interaction between job burnout and self-care (*β* = 0.146, *p* = 0.043) on retention intention were significant. Thus, job burnout could be interpreted to have a direct effect on retention intention, and self-care had a positive moderating effect on the effects of job burnout on retention intention (Table 6).

The final model for moderating the effects of self-care on the relationship between job burnout and retention intention derived from this study is shown in Figure 1.

## 4. Discussion

This study aimed to determine the importance of nurses’ self-care by assessing the moderating effects of self-care on the relationship between stress and retention intention and between job burnout and retention intention in clinical nurses.

First, perceived stress did not affect nurses’ retention intention. This result may be because the participants were asked about general stress, not job stress related to retention intention in the questionnaire. In other words, the questions did not ask about specific job-related stresses but rather included items such as daily conditions, irritation, and self-confidence; therefore, the possibility that other causes unrelated to the job were involved in the perceived stress might not be ruled out. The selection of the questionnaire could thus be considered a limitation in interpreting results. Therefore, a study that uses job stress rather than general perceived stress as a variable will be necessary.

It is worth discussing the findings in comparison to previous research. The perceived stress score of the participants in this study averaged 18.3 out of a maximum of 40. Higher scores indicate higher degrees of perceived stress. The average perceived stress score in this study was under the median, implying a low to moderate degree of perceived stress. This score was higher than the average score of 16.5 reported by Lim et al. [27] in a study that used the same instrument among emergency room nurses. Moreover, it was similar to the score of 18.8 found by Lee and Kim [28] in their study, conducted among clinical nurses. However, the average score in this study was lower compared to the study conducted by Park [29], which reported an average score of 22.7 among nurses in a regional trauma center. Among the studies using the same measurement tool, the highest perceived stress index was observed in the study conducted by Park [29], potentially due to the high severity of patient cases in the departments where the participants worked, resulting in higher levels of perceived stress.

Due to a lack of prior research that specifically focuses on perceived stress among clinical nurses, there are limitations preventing a simple comparison of the results. However, an examination of previous studies that investigated perceived stress variables among nurses shows that experiencing job stressors increases perceived stress [30], and depression, fatigue, and anger are associated with increased perceived stress among clinical nurses [28]. The consequences of the increased perceived stress may lead to decreased organizational productivity, reduced quantity and quality of nursing activities, decreased job satisfaction, and decreased job performance, all of which negatively impact patient care and individual nurses [12,31]. The practical impact of this stress is influenced by the subjective level of perceived stress as perceived by the individuals themselves, rather than an objectively measured degree of stress [32]. Therefore, perceived stress plays a crucial role in nursing practice. To reduce perceived stress, the implementation of stress management programs and activities is necessary.

The average degree of job burnout in this study was 2.61 out of 5, lower than the average of 2.90 in previous studies of nurses at tertiary general hospitals [33] and 3.14 in the average job burnout of studies involving oncology nurses [34]. Job burnout can be increased if a nurse is constantly placed in stressful working environments. In particular, nurses are more likely to suffer from burnout than other professionals because they constantly interact with other medical staff and patients and perform physical tasks, although they are in different work departments. This condition can negatively affect the nurses, the quality of nursing service, and even the performance of the organization [35]. Therefore, interventions should be developed to reduce the nurses’ job burnout and effectively respond to their problems. It is necessary to implement job burnout reduction programs that align with the specific context, environment, and culture of the hospital. In the United States, the mindfulness-based stress reduction program (MBSR) [36], reported to effectively reduce job stress and burnout, is actively used. Implementations of such programs may serve as a good example for reducing burnout.

The degree of self-care in this study was approximately 2.73 out of 5. A study on oncology nurses using the same self-care questionnaire as this study showed similar results to this study at 2.92 [15]. This average can be considered as representing a relatively moderate degree of self-care. As mentioned in the introduction, this result could be due to the lack of time devoted to self-care after work, as the number of nurses complaining about excessive workload and fatigue caused by a lack of nurses has increased.

The average retention intention in this study was 5.05 out of 8. In a previous study involving general nurses and using the same retention intention questionnaire as in this study, the degree of retention intention was 5.50 [37], which was higher than that in this study. This result might indicate that the changes in the medical environment are due to the effects of COVID-19 on retention intention [38]. Therefore, as the COVID-19 pandemic ends, efforts to improve the work environment suitably for the changed medical environment will be needed to improve retention intention.

Second, by comparing perceived general stress, job burnout, self-care, and retention intention according to the general characteristics of the study participants, such as gender, age, clinical work experience, marital status, religion, work department, and hobby status, significant differences were observed in perceived stress and job burnout by gender. In addition, significant differences were found in perceived stress, job burnout, and self-care by hobby status. Therefore, gender and hobbies should be considered when identifying intervention factors to reduce nurses’ perceived general stress and job burnout. In particular, perceived stress and job burnout were found to be higher among female nurses; thus, the causes of stress and job burnout and the factors influencing the differences based on gender should be identified. Hobbies were found to have a positive effect on reducing perceived stress and job burnout. However, individual preferences are different; therefore, providing support for hobbies in hospital systems is limited, and expanding opportunities for individuals to selectively participate by providing various hospital services should be considered.

Third, a high positive correlation was found between perceived general stress and job burnout. This finding is consistent with Lim’s [28] findings that the higher the perceived stress, the higher the likelihood of job burnout; therefore, the relationships among these two factors and their mutual influences should be considered rather than approaching each intervention separately to reduce perceived stress and job burnout. For this purpose, various aspects of analysis should be conducted, such as analyzing the detailed causes of stress and burnout. Job burnout and self-care showed a significant negative correlation, indicating that the former can be reduced through self-care activities; therefore, time for self-care activities or providing self-care services in hospitals could be provided. Similar to previous studies by Kim and Suh [39] and Moon [40], this study showed a significant negative correlation between job burnout and retention intention. In addition, self-care was significantly positively correlated with retention intention. Therefore, active support and program development at the organizational level should be promoted to reduce job burnout and improve retention intention.

Fourth, self-care was not found to be a significant moderator for the relationship between perceived stress and retention intention. This result may be because in the questionnaire, participants were asked about general stress, which includes individual life status, and not job stress related to retention intention. The experience of a specific event at work and the degree of general stress experienced by an individual in daily life may be different. This latter stress may include general stress experienced because of job-related stressors, job stress during an imbalance between an organization’s goals and the individual’s needs in the process of interacting within the organization, and the possibility of perceived results as a general stress situation in daily life.

Finally, self-care had a significant moderating effect on the relationship between job burnout and retention intention. No previous study has analyzed the effects of self-care on retention intention in nurses. Cho [41], who studied the relationship between self-care and job satisfaction in professional dancers, reported a significant correlation between self-care and job satisfaction. In other words, better self-care indicates a higher job satisfaction. A previous study also reported that job satisfaction strongly positively affects retention intention [42], which supports the findings of this study. Self-care is a factor that nurses can manage on their own even without organizational intervention; therefore, individuals should continue to perform self-care activities. In addition, at the organizational level, a supporting program for nurses that can motivate self-care activities should be considered to improve retention intention.

To explore organization-level strategies to reduce burnout among clinical nurses, previous research related to burnout was reviewed. According to a study on the effects of a resilience enhancement program for clinical nurses [43], resilience was negatively correlated with burnout. In other words, as resilience increased, burnout decreased. In that study, the resilience enhancement program provided to clinical nurses consisted of sessions aimed at improving stress coping abilities, interpersonal skills, positive belief, and maintaining resilience. Participation in this program was found to have a positive impact on reducing burnout among clinical nurses. Thus, it is necessary to consider the implementation of job burnout reduction programs (i.e., resilience enhancement programs) that align with the specific context and culture of the hospital.

Summarizing the above results, the lower the perceived stress, the lower the burnout, and the higher the self-care, the higher the retention intention. Although generalizing this study is difficult because it was a result of research on clinical nurses in one region, the factors affecting retention intention were identified using perceived stress and self-care as variables, which have not been covered in previous studies related to retention intention.

## 5. Conclusions

This study examined the effects of perceived stress, job burnout, and self-care on the retention intention of clinical nurses and identified the moderating effects of self-care on the relationship between job burnout and retention intention. In particular, improving retention intention can be achieved using the resources of each nurse (job burnout and self-care) in addition to the variables found in previous studies, such as job satisfaction, nursing environment, positive psychological capital, and nursing professionalism [24,44,45,46].

The study limitations are as follows. First, the selection of research participants was limited to specific hospitals in Seoul, restricting the generalizability of the findings. Second, there were limitations in using a general stress measurement questionnaire, which may have impacted the results. Therefore, future research should expand the study population, consider regional and hospital size variations, and compare and analyze the job retention intentions of advanced-beginner-stage nurses to generalize their findings. Lastly, it is necessary to utilize questions that investigate job-related stress.

Self-care can be managed by nurses themselves. Therefore, each nurse should identify their level of self-care activities and recognize the importance of self-care. To this end, it is necessary to consider creating an instructional program on the importance of self-care to conduct campaign activities at the organizational level. Moreover, different types of support should be provided at the organizational level to allow nurses to reduce their level of job burnout. Finally, no previous studies were found that were related to self-care in this context; thus, additional research should be conducted to verify the various effects of self-care.

## Figures and Tables

**Figure 1 healthcare-11-01870-f001:**
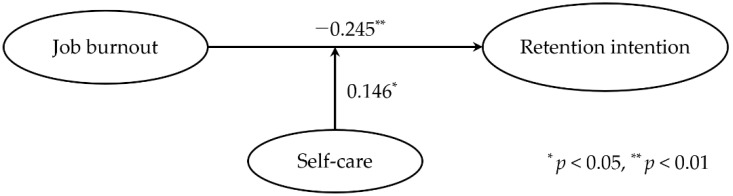
Final model for moderating the effects of self-care on the relationship between job burnout and retention intention.

**Table 1 healthcare-11-01870-t001:** General characteristics of the participants.

Characteristics	Category	*n* (%)
Gender	Female	157 (90.2)
	Male	17 (9.8)
Age(y)	Under 29	71 (40.8)
	30–34	52 (29.9)
	35–39	33 (19.0)
	Over 40	18 (10.3)
Clinical work experience	Beginner	8 (4.6)
	Advanced beginner	25 (14.4)
	Competent	51 (29.3)
	Proficient	90 (51.7)
Marital status	Single	128 (73.6)
	Married	46 (26.4)
Religion	Yes	63 (36.2)
	No	111 (63.8)
Department	Internal medicine ward	13 (7.5)
	Surgery ward	50 (28.7)
	Intensive care unit	48 (27.6)
	Emergency room	7 (4.0)
	Operating room	20 (11.5)
	Anesthesia recovery room	12 (6.9)
	Outpatient	24 (13.8)
Hobbies	Yes	99 (56.9)
	No	75 (43.1)

**Table 2 healthcare-11-01870-t002:** Descriptive analysis of variables.

Variables	Range	M ± SD	Skewness	Kurtosis
Perceived stress	0–4	1.83 ± 0.55	−0.07	−0.06
Job burnout	1–5	2.61 ± 0.62	0.11	−0.34
Self-care	1–5	2.73 ± 0.58	0.32	0.56
Retention intention	1–8	5.05 ± 1.66	−0.11	−0.80

**Table 3 healthcare-11-01870-t003:** Comparison of perceived stress, job burnout, self-care, and retention intention by general characteristics.

Characteristics	Category	Perceived Stress			Job Burnout			Self-Care			Retention Intention		
		M ± SD	t/F	*p*	M ± SD	t/F	*p*	M ± SD	t/F	*p*	M ± SD	t/F	*p*
Gender	Female	1.86 ± 0.53	2.34	0.020	2.66 ± 0.61	3.00	0.003	2.75 ± 0.58	1.60	0.111	4.99 ± 1.65	−1.48	0.140
	Male	1.54 ± 0.62			2.20 ± 0.54			2.52 ± 0.54			5.62 ± 1.68		
Age(y)	Under 29	1.86 ± 0.57	0.73	0.537	2.61 ± 0.63	0.41	0.746	2.79 ± 0.61	0.42	0.736	4.84 ± 1.58	0.96	0.414
	30–34	1.84 ± 0.56			2.68 ± 0.63			2.69 ± 0.47			5.12 ± 1.81		
	35–39	1.70 ± 0.54			2.57 ± 0.57			2.70 ± 0.65			5.15 ± 1.67		
	Over 40	1.88 ± 0.43			2.51 ± 0.64			2.73 ± 0.58			5.54 ± 1.47		
Clinical work experience	Beginner	1.70 ± 0.66	0.63	0.599	2.55 ± 0.58	0.69	0.557	2.82 ± 0.76	0.77	0.513	5.21 ± 1.80	0.74	0.532
	Advanced -beginner	1.94 ± 0.62			2.71 ± 0.65			2.81 ± 0.62			4.62 ± 1.52		
	Competent	1.84 ± 0.56			2.52 ± 0.55			2.79 ± 0.54			5.21 ± 1.71		
	Proficient	1.80 ± 0.51			2.64 ± 0.65			2.66 ± 0.58			5.07 ± 1.66		
Marital status	Single	1.82 ± 0.58	−0.30	0.765	2.63 ± 0.64	0.80	0.424	2.76 ± 0.59	1.23	0.221	4.98 ± 1.62	−0.98	0.331
	Married	1.85 ± 0.45			2.55 ± 0.54			2.64 ± 0.56			5.26 ± 1.76		
Religion	Yes	1.93 ± 0.54	1.86	0.065	2.67 ± 0.67	0.97	0.333	2.74 ± 0.59	0.13	0.897	5.16 ± 1.67	0.65	0.517
	No	1.77 ± 0.55			2.58 ± 0.59			2.72 ± 0.58			4.99 ± 1.66		
Department	Internal medicine ward	1.77 ± 0.66	0.44	0.851	2.63 ± 0.79	2.05	0.061	2.56 ± 0.81	1.42	0.211	5.55 ± 1.65	1.10	0.364
	Surgery ward	1.88 ± 0.56			2.71 ± 0.55			2.75 ± 0.56			5.00 ± 1.60		
	Intensive care unit	1.86 ± 0.61			2.70 ± 0.65			2.67 ± 0.57			5.31 ± 1.54		
	Emergency room	1.64 ± 0.63			2.07 ± 0.69			2.75 ± 0.62			5.64 ± 1.65		
	Operating room	1.82 ± 0.38			2.63 ± 0.51			2.55 ± 0.46			4.83 ± 1.66		
	Anesthesia recovery room	1.88 ± 0.56			2.61 ± 0.67			2.88 ± 0.56			4.39 ± 1.86		
	Outpatient	1.72 ± 0.45			2.36 ± 0.53			2.97 ± 0.56			4.71 ± 1.88		
Hobbies	Yes	1.75 ± 0.58	−2.08	0.039	2.53 ± 0.66	−1.99	0.048	2.93 ± 0.55	5.52	<0.001	5.02 ± 1.70	−0.33	0.744
	No	1.93 ± 0.49			2.72 ± 0.55			2.47 ± 0.52			5.10 ± 1.61		

**Table 4 healthcare-11-01870-t004:** Correlation between perceived stress, job burnout, self-care, and retention intention.

	Perceived Stress	Job Burnout	Self-Care	Retention Intention
Perceived stress	1			
Job burnout	0.587 **	1		
Self-care	−0.305 **	−0.287 **	1	
Retention intention	−0.272 **	−0.424 **	0.321 **	1

** *p* < 0.01.

**Table 5 healthcare-11-01870-t005:** Moderated multiple regression of hypothesis model.

Model	Variable	Unstandardized Coefficient		*β*	t	*p*	Tolerance	VIF	*R^2^*	Adj. *R^2^*	*R^2^* Change	F	*p*
		B	SE										
Level 1	Constant	6.917	0.773		8.944	<0.001			0.106	0.096	0.106	10.178	<0.001
	Perceived stress	−0.246	0.199	−0.095	−1.236	0.218	0.891	1.122					
	Job burnout	−0.485	0.108	−0.345	−4.504	<0.001	0.891	1.122					
Level 2	Constant	5.185	0.923		5.618	<0.001			0.158	0.144	0.052	10.667	0.001
	Perceived stress	−0.112	0.198	−0.043	−0.567	0.572	0.852	1.173					
	Job burnout	−0.363	0.111	−0.258	−3.260	0.001	0.789	1.267					
	Self-care	0.363	0.112	0.243	3.242	0.001	0.878	1.139					
Level 3	Constant	5.491	0.931		5.896	<0.001			0.183	0.158	0.024	7.515	0.085
	Perceived stress	−0.206	0.202	−0.079	−1.022	0.308	0.808	1.238					
	Job burnout	−0.395	0.113	−0.281	−3.511	0.001	0.758	1.319					
	Self-care	0.404	0.112	0.272	3.597	<0.001	0.854	1.171					
	Stress × Self-care	0.114	0.223	0.037	0.510	0.611	0.921	1.086					
	Burnout × Self-care	0.347	0.155	0.167	2.238	0.027	0.876	1.141					

**Table 6 healthcare-11-01870-t006:** Moderated multiple regression of the modified model.

Model	Variable	Unstandardized Coefficient		*β*	t	*p*	Tolerance	VIF	*R^2^*	Adj. *R^2^*	*R^2^* Change	F	*p*
		B	SE										
Level 1	Constant	6.023	0.273		22.045	<0.001			0.098	0.093	0.098	18.771	<0.001
	Job burnout	−0.441	0.102	−0.314	−4.333	<0.001	1.000	1.000					
Level 2	Constant	4.731	0.459		10.302	<0.001			0.157	0.147	0.058	15.902	0.001
	Job burnout	−0.339	0.103	−0.241	−3.293	0.001	0.918	1.089					
	Self-care	0.376	0.109	0.252	3.442	0.001	0.918	1.089					
Level 3	Constant	4.654	0.457		10.192	<0.001			0.177	0.163	0.020	12.191	0.043
	Job burnout	−0.345	0.102	−0.245	−3.373	0.001	0.917	1.090					
	Self-care	0.420	0.110	0.282	3.810	<0.001	0.882	1.134					
	Burnout × Self-care	0.303	0.148	0.146	2.044	0.043	0.954	1.048					

## Data Availability

The research data can be requested from the first author.

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
