# Peer review of "The Moderating Effects of Self-Care on the Relationships between Perceived Stress, Job Burnout and Retention Intention in Clinical Nurses"

_healthcare, 2023, doi:10.3390/healthcare11131870_

Round 1
Reviewer 1 Report
Dear author
I would like to make a few comments below that I think need to be clarified.
Improve the mesh used (e.g. use "Retention, Psychology", instead of "Retention" exclusively).
In the summary, make explicit the instruments used for the determination of stress and burnout, as well as the statistical tests used.
Check the references used. More than 50% are prior to 2017, and of these 50% are prior to 2013. Please include more up-to-date references.
Please review the presentation format. e.g. review the characters in bold format.
A descriptive study should not contain hypothesis formulation. Please correct this. If you want to make hypotheses, you should vary the nomenclature of the study. Remember that descriptive studies aim to collect data on a phenomenon in order to "describe" what is being investigated. It is not intended to analyse causal relationships; this is typical of correlational designs, not descriptive designs.
Do not formulate hypotheses, use the conventions; research hypotheses; alternative hypotheses, etc. As well as the statistical values of acceptance of the hypothesis.
Please detail the characteristics of the participating hospitals, as well as the characteristics of the participating nurses' units; as perceived stress in one unit may not necessarily be the same as perceived stress in other units; for example, in an intensive care unit, perceived stress may be higher than in a maternity unit. For example, in your study, the sample belonging to intensive care, emergency and anaesthesia units accounts for almost 40% of the interviewed population.
Therefore, please describe and justify this aspect adequately.
Justify the type of sampling you used and the sample size calculation; whether you used probability cluster sampling, or whether you randomly sampled participants without regard to the facility they were in, etc.
The statistics used for correlations (T-Test) are used for normal population. Justify their use. You do not provide us with anything in relation to the normality characteristics of your population. Therefore, your population is statistically normal? justify it conveniently. If it is, the analysis is adequate; if it is not, use another test appropriate to the characteristics of your population and the variables to be analysed.
Author Response
Thanks for your feedback and comments. We revised the manuscript as you suggested. Please find the 'response to reviewer' document.

Reviewer 2 Report
To better understand the current nursing issues of burnout and turnover, the authors examined the possible moderating effect of self-care on the relationship between job stress and retention intention as well as the relationship between job burnout and retention intention. The study provides a useful contribution in that it examines the impact of the nurse’s personal actions (engaging in self-care) on reducing the impact of overall personal stress and burnout on the work/organizational outcome of turnover.
While there is no explicit overall theory to guide the study, the introduction includes a brief overview of the reasoning used for each path/relationship noted in the study model (figure 1). A few points of clarification are suggested.
-lines 33-36 – This is a run-on sentence which is convoluted. Suggest dividing the statement “…patient’s healthy life and death. In turn, resilience is reduced and, if job stress continues, burnout occurs.”-line 43 – it is not clear what is meant by ‘in-service nurses’, suggest using another term.
-line 54 – The authors claim that ‘no study directly addressed’ the study variables but no description of the literature search was provided. It might be more accurate to state that ‘no studies were located’ meaning that they could exist but the authors did not find them in the methods used for the literature search.
-line 61-62 – the following statement is not clear ‘studies of realistic and applicable measures are still lacking’.
-line 74-75 – The final study purpose statement needs to be more specific to reflect the study model provided. Suggest edit, ‘Investigating the moderating effects of self-care on the relationship between stress and retention intention as well as on the relationship between job burnout and retention intention.’ A similar edit is then needed for line 88.
The Design and methods were appropriate for the study purpose. An online survey was completed by nurses employed in three hospitals in Seoul, South Korea. Sample size was justified. It would be useful to also add the inclusion/exclusion criteria and the sampling method that was used so that the overall response rate could be calculated and reported. Each instrument used in the survey was described briefly. It would be useful to have an example item included for each instrument. Further description of the Likert scale anchors for burnout, self-care and retention intention is advised (e.g., 0=never, etc.). Suggested edits are as follows:
-lines 114-117 – To be more succinct, suggest revising the statement to ‘The reliability of K_BAT was within acceptable limits with the Cronbach alpha ranging from .90-.92 for the subfactors. In this study, the Cronbach alpha ranged from .86-.91 across subfactors with an overall reliability of .91.’. I suggest a similar edit when describing the self-care instrument (lines 132-138) and introducing the discussion (lines 233-34).
-line 123 – Suggested revision ‘reverse-ordered questions’.
-line 141 – To be more specific, suggested revising to ‘Ethics approval was received by the Institutional Review Board…’
-line 144 – Suggested revision ‘a recruitment notice was posted…’
The results section, statistical analysis and tables were described well. It would be useful to include highlights of the description of the sample based on the details provided in Table 1. An overall mean and standard deviation for each study model variable is also recommended. Suggested edits are as follows:
-line 172 – The following phrase is unclear ‘their intention to use was compared’.
-line 184 – Table 2 – for each study model variable, add the scale range (in the column header) so that the values provided can be interpreted by the reader.
-line 192 – In Table 3, remove the indent for burnout and self-care.
-line 229 – Figure 2 – suggest adding in the beta coefficients and significance level for each path.
The discussion of findings was thorough. The main flaw that the authors acknowledge is the lack of consistency between the conceptual definition of job stress and operational definition as the instrument selected measured general stress rather than job stress. Therefore, the discussion on stress should be focused on general stress for the remainder of this section. Clarifications, edits and suggestions are offered as follows:
-line 242 – suggest using ‘interpreting results’ instead of ‘deriving results’. It would be useful to comment on the level of stress that the nurses reported i.e., how would you interpret the mean stress of 1.54-1.94 on a scale from 0-4 (from Table 2)? Is this low, moderate, high and how does it compare to other studies of a similar population? This also applies to each study model variable.
-line 244 – a burnout score of 2.61 was reported. Please expand this and report it as a mean and include the standard deviation. This applies to each study model variable.
-line 249 – not sure of the intended meaning of ‘a different working environment’.
-line 251 – suggested edit, from ‘medical service’ to ‘nursing service’.
-line 253-56 – Suggest reversing the order of these 2 sentences i.e., mention the need for a burnout intervention then offer an example of the MBSR. In your recommendation to offer an intervention, you state ‘according to the national medical system’. It is not clear why this was added. Perhaps you are meaning that the intervention needs to match the specific hospital context and culture.
-line 260 – a mean for self-care of 2.73/5 is moderate and not low as you suggest in this statement.
-line 277-283 – To be more specific, suggest using ‘general stress’ each time stress is mentioned in the remainder of the discussion since that is concept that was being measured (as opposed to job stress). As for recommendations, I don’t think the hospital is responsible for personal stress or hobbies. Alternatively, they can recommend that nurses engage in hobbies to reduce personal stress.
-line 290-297 –Suggest being more tentative about your recommendation (use ‘could’ instead of ‘should’) since burnout and self-care were negatively correlated (mall correlation) but causation was not tested. I think it would be helpful to consider what obligations the hospital has for self-care since this is an individual nurse responsibility. Instead, you might want to consider reducing the root causes of burnout that are within the responsibility of the organization. In other words, treat the cause and not the symptom.
-line 298-299 – Awkward sentence. Suggested revision ‘Self-care was not found to be a significant moderator for the relationship between personal stress and retention intention.’
-lines 308-311 – As above, suggested revision ‘Self-care had a significant moderating effect on the relationship between job burnout and retention intention.’ Delete other existing statements and then continue with ’No previous study has analyzed the effects of self-care on retention intention in nurses. Cho [37], who studied….’.
-line 318 – Suggest replacing ‘mediate’ with ‘manage’. For this paragraph, you might want to comment on the size of the moderating effect of self-care so that the recommendations are proportionate.
-line 322 – Suggest replacing ‘better’ with ‘higher’.
-line 326-328 – Awkward sentence. Suggest deleting it as it is repetitive of the previous points.
For the conclusion, I have some concerns about the tone that suggests the nurse is responsible for reducing their own burnout through self-care. I think this misses the responsibility of the organization to reduce the root causes of burnout and may overstate the moderation effect of self-care on the relationship between burnout and retention intention. The paragraph could be revised to indicate that nurses can have some level of control over their burnout experience so that it doesn’t result in leaving their job but organizations must play a part in reducing the causes of burnout.
-line 331-332 – Need to be more specific, suggested edit ‘…identified the moderating effects of self-care on the relationship between job burnout and retention intention.’
-line 337 – Replace ‘mediate’ with ‘manage’.
-line 342- Suggested edit ‘self-care studies were not found’ instead of ‘cannot be found’
Author Response

(The authors gave the same response as above.)

Reviewer 3 Report
Good job with the choice of topic which focuses on nurses' self care and retention. These two are very relevant in our healthcare system, worldwide. However, there are rooms for improvement regarding this article.
Your abstract did not reflect the type of study you embarked on, how your participants were selected, and how data collection was completed. Your article should reflect these so that readers, at the glimpse, should know what you have done.
Under Introduction, you used an abbreviation that is NOT universal. What does OECD in line 29 stand for? Before using abbreviation, it is advisable to write the full name followed by abbreviation.
Theoretical or conceptual frameworks are vital in studies. What theoretical or conceptual framework did you use for this study? This should have served as guide for your hypotheses. I did not come across any in your article.
Under Research Methods, you did not provide information on the settings other than being three "large hospitals". Are they all similar in terms of being all public hospitals, teaching, or private hospitals?
How many participants came from each of the hospitals?
The article should reflect the type of study design you embarked on.
Under Results, from Line 185 to 193 including Table 3., the figure you reported in your writing is different from what is on the table with regards to the correlation between perceived stress and burn out. Check your figures again.
Any limitations beside using 'moderated multiple regression?' Between your Discussion and Conclusion, you should indicate limitations of this study.
I hope you take time to revisit your article and make revisions because this is an important topic for nurses.
Good luck
There are flaws with sentence structure and grammar. One such example can be found from lines 33 to 36. Consider restructuring this so the the readers would be able to follow. In line 43, what do you mean by " other in-service nurses"? I recommend that you engage the services of an English Language Editor to assist with revising your article.
Author Response

(The authors gave the same response as above.)

Round 2
Reviewer 1 Report
Dear Autor.
Thank you for your work and consider the recommendations given.
I consider the modifications made to be sufficient.
However, there are still several references that are too old (1996, 2002, 2004, etc). Could they be replaced by other references that are more up to date?
Author Response
Thanks for your comment. Please find the attached.
